# Meta-analysis of the correlation between serum uric acid level and carotid intima-media thickness

**Mingzhu Ma**[1]ᵒ, **Liangxu Wang**[2]ᵒ, **Wenjing Huang**[1], **Xiaoni Zhong**[1], **Longfei Li**[1], **Huan Wang**[3], **Bin Peng**[1], **Min Mao**[4]*

**1** School of Public Health and Management, Chongqing Medical University; Research Center for Medicine and Social Development, Chongqing, China, **2** School of Basic Medicine, Kunming Medical University, Kunming, China, **3** Department of Development planning, Chongqing Medical University, Chongqing, China, **4** Department of Cardiology, The First Affiliated Hospital of Chongqing Medical University, Chongqing, China

ᵒ These authors contributed equally to this work.
* 187169431@qq.com

## Abstract

### Objective

Recently, increasing epidemiological evidence has shown that there is a correlation between serum uric acid level (SUA) and carotid intima-media thickness (CIMT). This paper explored the relationship between them through meta-analysis.

### Methods

PubMed, Cochrane Library, EMBASE, Web of Science and Google Scholar were searched to obtain literature. The keywords used to retrieve the literature were carotid intima thickness, intima-media thickness, carotid atherosclerosis, carotid stenosis, carotid artery, uric acid, blood uric acid, and hyperuricaemia. The retrieval time was from the establishment of the database through July 2020. Stata15.0 and RevMan5.3 software were used for statistical analysis. The standardized mean difference (SMD) and 95% confidence interval (95% CI) were calculated by a random effect model to estimate the correlation. Publication bias was assessed using the Begg and Egger tests. The stability of these results was evaluated using sensitivity analyses.

### Results

Fifteen studies were included with a total sample size of 11382, including 7597 participants in the high uric acid group and 3785 in the control group, on the basis of the inclusion and exclusion criteria. According to the evaluation of the JBI scale, the literature was of high quality. The average age ranged from 42 to 74. Meta-analysis showed that CIMT in the high uric acid group was significantly higher than that in the control group (SMD = 0.53, 95% CI: [0.38, 0.68]), and the difference was significant (z = 6.98, $P < 0.00001$). The heterogeneity among the 15 articles was obvious ($I^2 = 89\%$, $P < 0.00001$). Subgroup analysis by disease status illustrated a positive relationship between SUA and CIMT in healthy people and

**Data Availability Statement:** All relevant data are within the manuscript and its Supporting Information files.

**Funding:** This work was supported by the Chongqing Science and Technology Commission (cstc2018jscx-msybX003) and Chongqing Municipal Health and Health Committee (ZY201802121). The funders had no role in study design, data collection and analysis, decision to publish, or preparation of the manuscript. The authors received no specific funding for this work.

**Competing interests:** The authors have declared that no competing interests exist.

people with diseases. SUA was shown to be positively correlated with CIMT in people aged 45–60 years and ≥60 years by subgroup analysis by age. SUA was also found to be positively correlated with CIMT in a population with BMI>24 kg/m$^2$ by subgroup analysis by BMI. In addition, subgroup analysis of other risk factors for CIMT, including TC, SBP, DBP, triglycerides, and LDL-C, all showed a positive correlation between SUA and CIMT.

## Conclusions

There is a significant correlation between serum uric acid level and carotid intima-media thickness, and a high concentration of serum uric acid is related to carotid artery intima-media thickness.

## Introduction

Hyperuricaemia is a disorder caused by the dysfunction of purine metabolism. When the level of uric acid in the human body is imbalanced, synthesis is reduced, and hyperuricaemia occurs. Hyperuricaemia affects blood vessels, limbs and joints and damages many important organs in the human body, and it is an important biochemical basis of gout. Gout can lead to kidney stones, uric acid nephropathy and even renal failure. With substantial changes in the traditional lifestyle, the consumption of high purine and high protein foods has increased, and hyperuricaemia has become a common clinical disease. The incidence of hyperuricaemia has shown a significant upward trend in recent years [1–3]. The prevalence of hyperuricaemia and gout in mainland China from 2000 to 2014 was 13.3% and 1.1%, respectively [4]. In a nationally representative sample of U.S. adults, the prevalence of gout and hyperuricaemia remained high from 2007 to 2016. In 2015–2016, the prevalence of gout in adults in the United States was 3.9% (9.2 million), with 5.2% (5.9 million) in men and 2.7% (3.3 million) in women. The prevalence of hyperuricaemia was 20.2% and 20.0%, respectively [5].

The prevalence of atherosclerotic cardiovascular disease (ASCVD) is increasing and has become the leading cause of death and disability worldwide [6], with significant negative impacts on individuals, families and societies. Both the heart and brain exhibit similar vascular anatomy, with large ductal arteries extending from the surface of the organs and providing tissue perfusion through a complex network of small vessels. Both organs rely on fine-tuning of local blood flow to meet metabolic demands [7]. Atherosclerosis is the common pathological basis of cardiovascular and cerebrovascular diseases, and there is a strong correlation between carotid atherosclerosis and cardiovascular and cerebrovascular diseases. Carotid intimal thickening is considered to be an objective indicator of early atherosclerosis and an effective predictor of systemic atherosclerosis, which is closely related to the occurrence of coronary heart disease and ischaemic stroke [8].

At present, the relationship between serum uric acid level and carotid intima-media thickness has become a research hotspot, but the research results are controversial. A number of studies have found that the serum uric acid level is independently related to CIMT thickening [9, 10], but some scholars have concluded that there is no significant correlation between SUA level and IMT values in men or women [11–14]. These studies indicated that whether serum uric acid levels can be considered an independent risk factor for carotid atherosclerosis remains highly controversial [15]. In conclusion, the relationship between serum uric acid levels and carotid intimal thickness is not completely clear, and there is a lack of large-sample comprehensive and systematic research. Therefore, we carried out a meta-analysis through a

series of literature searches and performed a comprehensive evaluation of the relationship between the two factors by using standardized mean difference (SMD) to provide supporting evidence for further related research.

## Materials and methods

### Literature retrieval strategy

According to the guidelines of Preferred Reporting Items in Systematic Reviews and Meta-Analyses (PRISMA) (moher 113 et al. 2015 [16]), PubMed, Cochrane Library, EMBASE, Google Scholar and Web of Science were searched. The retrieval time was from the establishment of the database through July 2020. The following keywords were used in the retrieval process: carotid intima thickness, intima-media thickness, carotid atherosclerosis, carotid stenosis, carotid artery, uric acid, blood uric acid, and hyperuricaemia. There were no restrictions on language or region. To prevent the omission of retrieval, the titles and abstracts of the references were read and screened.

### Inclusion and exclusion criteria

According to the purpose of this study, first, all published studies on the relationship between serum uric acid level and carotid intimal thickness were included, and the following inclusion and exclusion criteria were used. The inclusion criteria were (1) specific sample size values; (2) complete demographic data; (3) division of subjects into a high uric acid group and control group according to uric acid level; and (4) measurement of the CIMT. The exclusion criteria were (1) no control group; (2) missing data; (3) no specific CIMT value; (4) duplicate publication; and (5) lectures and review articles.

### Data extraction and quality assessment

Literature inclusion and data extraction were conducted independently by two researchers (Mingzhu Ma and Liangxu Wang). When the results were inconsistent, experienced researchers were consulted to discuss the final results. The data extraction table includes the following contents: (1) basic characteristics of the article, including author, year, region, average age of research population, and confounding factors; and (2) effect-related data, such as sample size, mean value, and standard deviation of the high uric acid group and control group. The JBI scale was used to evaluate the quality of the studies. The scale was developed by the Joanna Evidence-based Nursing Center (JBI) in Australia. There are 10 items in the scale, and the evaluation criteria are divided into three levels: 0 points–not in accordance with the standard; 1 point–mentioned but did not provide a detailed description; and 2 points–provided a detailed, comprehensive and correct description. Articles with scores greater than 14 can be regarded as high-quality research.

### Statistical analysis

Due to differences in research design between studies, the data extracted from the original studies were appropriately transformed before analysis [17]: (1) for studies that classified the participants into a high uric acid group and a control group, the extracted data were used for estimation without conversion; (2) for studies dividing the participants into multiple groups (or more than two groups), according to the definition criteria of hyperuricaemia [18], individuals meeting these standards were categorised into the high uric acid group, and the rest composed the control group; (3) hyperuricaemia was defined as > 7 mg/dl (416 μmol/L) for men and > 6 mg/dl for women, and the included studies were not grouped according to sex.

We adopted the lower female criteria ($> 6$ mg/dl) as the cut-off for grouping. The calculation method of the mean value and standard deviation after combination is as follows [19]: supposing that the mean value and standard deviation of each group before combination are $\bar{X}_T$ and SD$_T$, respectively, then the combined mean and standard deviation are $\bar{x_T} = \dfrac{\sum_{i=1}^{m} n_i \bar{x}_i}{\sum_{i=1}^{m} n_i}$ and

$SD_T = \sqrt{\dfrac{\sum_{i=1}^{m}(n_i - 1)SD_i^2 + \sum_{i=1}^{m} n_i(\bar{x}_i - \bar{x_T})^2}{(\sum_{i=1}^{m} n_i - 1)}}$. Since the measurement methods and units of CIMT in different studies are not exactly the same, the effect value SMD was used to eliminate the influence of differences. The random effects model was selected in the analysis. Heterogeneity between studies was assessed using forest plots, Q tests, and $I^2$. The critical value of the Q-test was set at 0.1, and heterogeneity was considered obvious when $I^2 > 50\%$. Publication bias was initially judged by forest plot symmetry, and then the Begg rank correlation test and Egger's test were performed. There was no publication bias when $z < 1.96$, $P > 0.05$ in the Begg test, and $P > 0.05$ with the 95% CI including 0 in the Egger test. Sensitivity analysis was used to determine the individual impact of each study on the combined results. Stata15.0 and RevMan5.3 software were used for the above analysis, and the literature quality score figure was created with R3.6.3.

## Results

### Literature screening process

A total of 453 studies were initially retrieved (including those identified by browsing the references of the preliminarily screened literature by the literature tracking method to make up for the possible omission of computer retrieval). Duplicate publications, lectures and review articles (n = 58), as well as articles that were not relevant to the subject and did not meet the inclusion criteria (n = 256), were excluded. The remaining 139 studies were screened after reading the full text, and 15 studies were included in the meta-analysis [20–34]. The specific process is shown in Fig 1.

### Basic characteristics and quality evaluation of the literature

A total of 15 articles were included in this meta-analysis, with a total sample size of 11382, including 7597 participants in the high uric acid group and 3785 participants in the control group. The study with the largest sample size included 5294 subjects (Zhang Hailing, 2016, Hebei, China), and the smallest included 80 (Yusuf Tavil, 2017, Turkey). The average age ranged from 42 to 74. In addition to healthy middle-aged and elderly people, the study population included people with type 2 diabetes, ischaemic stroke, hypertension, renal transplant, cardiac syndrome X and metabolic syndrome. The basic characteristics of the literature are shown in Table 1, and the detailed characteristics are shown in S1 Table. The quality scores of each study were evaluated by two researchers according to the 10 items of the JBI scale. When there were inconsistencies, experienced researchers were consulted. After discussion and agreement, the final data were entered. The results showed that the total score of all studies was 14 or above, and the quality of the literature was high (see Fig 2).

### Meta-analysis results

The results of the combined analysis of the random effect model showed that the CIMT value of the high uric acid group was significantly higher than that of the control group (SMD = 0.50, 95% CI = [0.34, 0.66]), and the difference was statistically significant (z = 6.24,

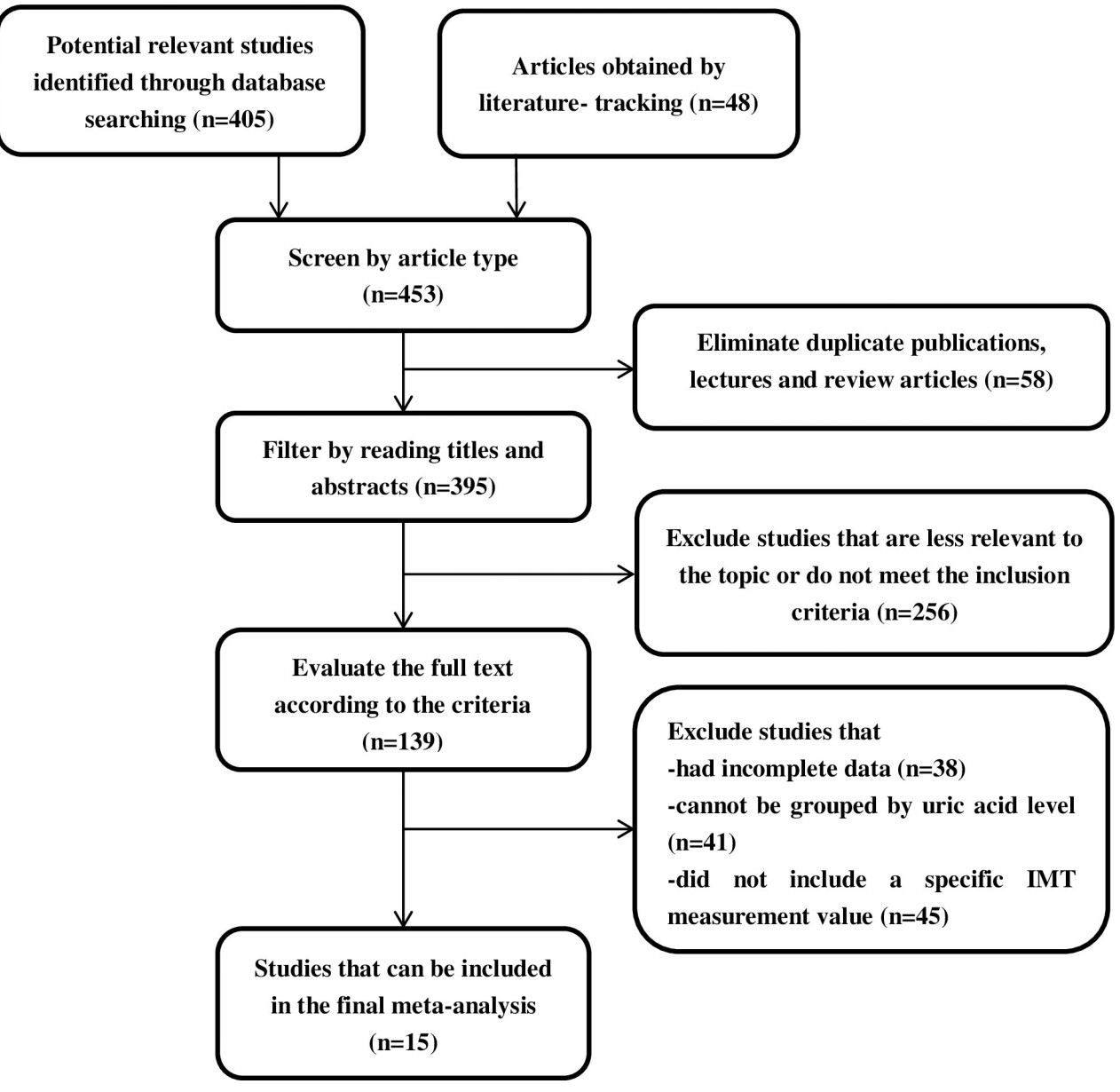

**Fig 1. Literature screening process.**

$P < 0.00001$) (Table 2). There was significant heterogeneity among the 15 included studies ($I^2 = 89\%$, $P < 0.00001$) (Table 3). The forest plot is shown in Fig 3.

Because the correlation between SUA and CIMT is affected by population characteristics and may be affected by other risk factors for CIMT, we conducted subgroup analyses. Initially, we divided the population into two groups and found that CIMT levels were higher in the high uric acid group than in the control group, both in healthy people and in people with diseases. (Healthy people: SMD = 0.29, 95% CI: [0.17, 0.42]; people with diseases: SMD = 0.73, 95% CI: [0.52, 0.95]). The differences were statistically significant ($P < 0.00001$), as shown in Fig 4, which meant that SUA levels were positively correlated with CIMT. In addition, the heterogeneity of the two subgroups decreased but was still significant ($I^2 = 78\%$, $P = 0.0001$; $I^2 = 73\%$, $P = 0.0005$).

**Table 1. Basic characteristics of included literatures.**

| N | Author | Year | Region | Sample size | | CIMT(mm) | | Average age | Population prevalence |
|---|---|---|---|---|---|---|---|---|---|
| | | | | High uric acid group | Control group | High uric acid group | Control group | | |
| 1 | Ranran Zhang[20] | 2018 | Qingdao, China | 234 | 100 | 1.06±0.26 | 1.01±0.24 | 59.5 | |
| 2 | Hailing Zhang[21] | 2016 | Hebei, China | 4739 | 555 | 0.93±0.19 | 0.84±0.19 | 55.0 | |
| 3 | Zhigai Zhang[22] | 2013 | Henan, China | 54 | 54 | 1.05±0.28 | 0.84±0.19 | 52.5 | Type 2 diabetes |
| 4 | Chunyu Yang[23] | 2005 | Guangdong, China | 41 | 67 | 0.99±0.18 | 0.94±0.17 | – | |
| 5 | Francesco Antonini-Canterin[24] | 2019 | Italy | 136 | 562 | 0.97±0.22 | 0.94±0.18 | 57.3 | |
| 6 | Qin Li[25] | 2011 | Shanghai, China | 513 | 513 | 0.895±0.27 | 0.94±0.22 | 65.6 | Type 2 diabetes |
| 7 | Chun-Chin Chang[26] | 2018 | Taiwan, China | 660 | 313 | 0.675±0.135 | 0.94±0.11 | 61.7 | |
| 8 | Kumral.E[27] | 2013 | Turkey | 75 | 331 | 0.9±0.3 | 0.94±0.2 | 66.0 | Ischemic stroke |
| 9 | Young Seok Cho[13] | 2018 | Korean | 505 | 465 | 0.66±0.14 | 0.65±0.14 | 52.7 | |
| 10 | Ryuichi Kawamoto[28] | 2005 | Japan | 308 | 611 | 1.07±0.23 | 1.00±0.22 | 74.3 | |
| 11 | Mustafa Caliskan[29] | 2014 | Turkey | 114 | 38 | 0.58±0.09 | 0.52±0.09 | 44.6 | Masked hypertension |
| 12 | E.Asicioglu[30] | 2014 | Turkey | 46 | 44 | 0.58±0.09 | 0.57±0.1 | 41.5 | Renal Transplant |
| 13 | Nusret Acikgoz[31] | 2011 | Turkey | 50 | 40 | 0.75±0.18 | 0.63±0.09 | 51.8 | Cardiac Syndrome X |
| 14 | Yusuf Tavil[32] | 2007 | Turkey | 55 | 25 | 0.76±0.15 | 0.57±0.16 | 50.3 | Hypertension |
| 15 | Shun-Sheng Wu[33] | 2019 | Taiwan, China | 67 | 67 | 0.75±0.11 | 0.66±0.1 | 50.8 | Metabolic Syndrome |

To further control for confounders, we performed an age-based subgroup analysis and found a positive correlation between SUA and CIMT in people aged 45–60 years and $\geq$60 years (Fig 4), but no correlation was found in people between the ages of 18 and 45. We also performed a subgroup analysis based on BMI. The SUA level was significantly positively correlated with CIMT in people with BMI>24 kg/m$^2$. In addition, we performed subgroup analysis on other risk factors for CIMT: TC, SBP, DBP, triglycerides and LDL-C (the forest plots are shown in S1 Appendix). The results showed that the carotid intimal thickness of the high uric acid group was higher than that of the control group, and the difference was statistically significant (Table 2). The heterogeneity of each subgroup is shown in Table 3.

## Publication bias and sensitivity analysis

To measure publication bias, a forest plot was drawn. The distribution on both sides of the graph is approximately symmetrical (Fig 5). The results of Begg's and Egger's tests showed that there was no significant publication bias among the articles included in this meta-analysis. (Begg: z = 1.39, $P$ = 0.166; Egger: $P$ = 0.530, 95% CI: [- 2.450, 4.536]). One study at a time was removed from the total combined results to explore the impact of the study on the overall SMD. The results showed that the combined values of the remaining research exhibited no significant differences compared with the original combined values (Fig 6). The sensitivity analysis showed the stability of the results.

## Discussion

Carotid intima-media thickness is associated with the degree of coronary atherosclerosis and the outcome of ischaemic stroke and has been used as an alternative marker for cardiovascular and cerebrovascular diseases in clinical trials and observational studies [34–37]. The

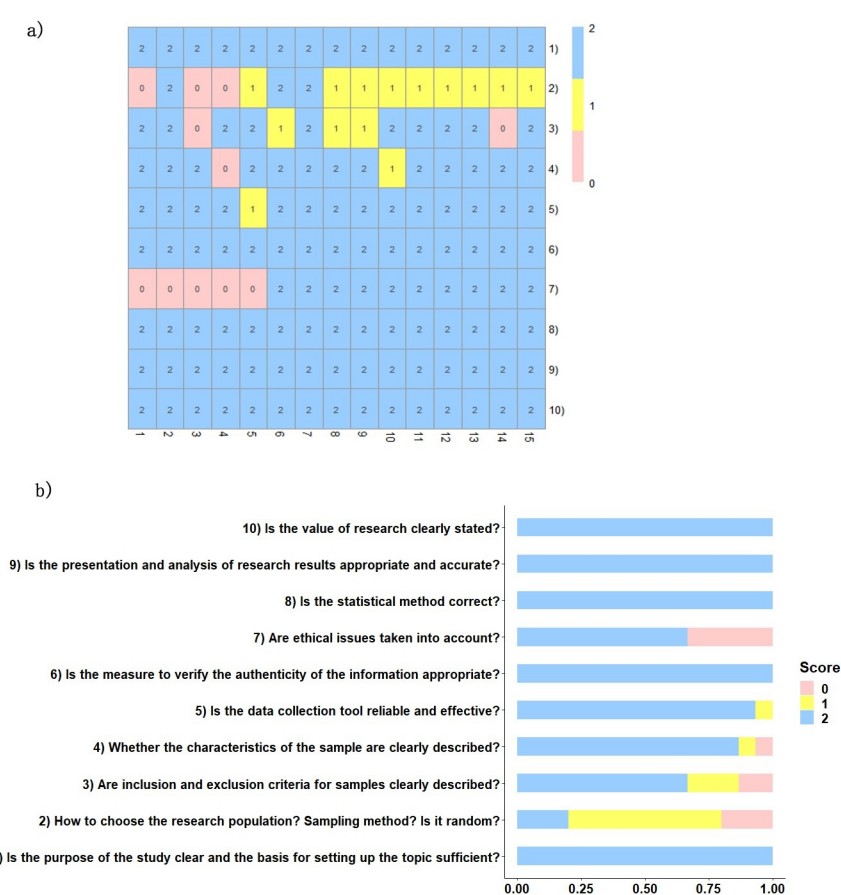

**Fig 2. Quality score of literature.** (a) The thermal map of quality scores: 1–15 represents the 15 studies included, 1)-10) represents the 10 evaluation items. (b) Bar chart of the score distribution of each item.

destruction of the vascular wall and endothelial cell function are direct causes of atherosclerosis. As an end product of purine metabolism in humans and higher primates, uric acid can damage the integrity of the vascular surface and affect the function of endothelial cells through direct or indirect effects on the vascular endothelium. The mechanism of this process is as follows: uric acid can promote the proliferation and migration of vascular smooth muscle cells and even lead to vascular remodelling, affecting its normal function. When the concentration of uric acid in the blood reaches a certain level, it will deposit into the tissue in the form of urate crystals and damage the vascular endothelium. The occurrence of hyperuricaemia aggravates abnormalities in metabolites and affects the vascular endothelium and haemodynamics [20]. Oxidative stress is the key link in endothelial dysfunction caused by hyperuricaemia and is related to the increased risk of cardiovascular and cerebrovascular disease [38].

Because uric acid is generally considered to have more harmful than protective effects, it is considered to be a risk factor for CIMT and atherosclerosis [35, 36, 33], but the clinical relevance of the two has been controversial. Since 1950, research on the relationship between serum uric acid level and carotid intima-media thickness has attracted scholars' attention [39]. In a family-based study, 449 members of 107 families were analysed, and the results showed that SUA plays an important regulatory role in atherosclerosis [40]. Another cross-sectional study of 8144 healthy people reached the same conclusion and noted that SUA is an independent risk factor for atherosclerotic plaque [41]. In contrast, some studies have shown that

**Table 2. Overall and subgroup analysis of the correlation between SUA and CIMT.**

| | Number | SMD | 95% CI | z | P |
|---|---|---|---|---|---|
| **Overall** | 15 | 0.50 | [0.34,0.66] | 6.24 | <0.00001 |
| **Population prevalence** | | | | | |
| Healthy people | 7 | 0.29 | [0.17,0.42] | 4.55 | <0.00001 |
| People with diseases | 8 | 0.73 | [0.52,0.95] | 6.59 | <0.00001 |
| **Age** | | | | | |
| 18–45 | 2 | 0.39 | [-0.16,0.94] | 1.4 | 0.16 |
| 45–60 | 8 | 0.53 | [0.32,0.75] | 4.85 | <0.00001 |
| ≥60 | 4 | 0.52 | [0.19,0.84] | 3.09 | 0.002 |
| **BMI (kg/m$^2$)** | | | | | |
| ≤24 | 2 | 0.74 | [-0.16,1.63] | 1.61 | 0.11 |
| >24 | 8 | 0.52 | [0.28,0.76] | 4.29 | <0.0001 |
| **TC (mmol/L)** | | | | | |
| <5.2 | 7 | 0.45 | [0.29,0.61] | 5.57 | <0.00001 |
| ≥5.2 | 5 | 0.70 | [0.37,1.04] | 4.09 | <0.0001 |
| **SBP (mmhg)** | | | | | |
| <140 | 7 | 0.44 | [0.27,0.61] | 5.10 | <0.00001 |
| ≥140 | 3 | 0.80 | [0.29,1.30] | 3.08 | 0.002 |
| **DBP (mmhg)** | | | | | |
| <90 | 8 | 0.53 | [0.30,0.76] | 4.57 | <0.00001 |
| ≥90 | 1 | 1.23 | [0.72,1.74] | 4.71 | <0.00001 |
| **Triglyceride(mmol/L)** | | | | | |
| <1.7 | 9 | 0.46 | [0.28,0.64] | 4.98 | <0.00001 |
| ≥1.7 | 1 | 0.94 | [0.81,1.07] | 14.29 | <0.00001 |
| **LDL-C(mmol/L)** | | | | | |
| <3.4 | 8 | 0.51 | [0.25,0.76] | 3.84 | 0.0001 |
| ≥3.4 | 1 | 1.23 | [0.72,1.74] | 4.71 | <0.00001 |

Note: (a) SMD, standardized mean difference; 95% CI, 95% confidence interval; z, significance test for SMD = 0; (b) TC, total cholesterol; SBP, systolic blood pressure; DBP, diastolic blood pressure; HDL-C, high-density lipoprotein cholesterol; LDL-C, low-density lipoprotein cholesterol.

serum uric acid levels are not an independent risk factor for cardiovascular and cerebrovascular diseases and atherosclerosis. A cohort study based on rural communities found that serum uric acid level was a risk factor for arterial stiffness but was not associated with carotid intimal thickness [12].

As the level of serum uric acid in adults will be affected by many factors, other risk factors for cardiovascular and cerebrovascular diseases, such as hypertension, diabetes, metabolic syndrome, smoking, alcohol consumption, etc., will have an impact on it [26]. In people with multiple diseases, the relationship between SUA and atherosclerosis is likely to be influenced by other cardiovascular and cerebrovascular risk factors [42]. In previous studies, when clarifying the role of serum uric acid levels in cardiovascular disease, the results were greatly limited by the single research population [26], which made it difficult to explain this debate. Therefore, it is important to carry out systematic large-sample studies involving various populations.

In this paper, we systematically reviewed the published studies on the relationship between uric acid concentration and carotid intimal thickness through a meta-analysis. In addition to healthy people, the study population included participants with type 2 diabetes, hypertension, ischaemic stroke, renal transplant and metabolic syndrome. Meta-analysis showed that there was a significant correlation between SUA and CIMT. To exclude the deviation caused by

**Table 3. Heterogeneity test of the included literature and subgroups.**

|  | Number | Chi$^2$ | P | I$^2$ |
|---|---|---|---|---|
| **All literature** | 15 | 128.08 | <0.00001 | 89% |
| **Population prevalence** |  |  |  |  |
| Healthy people | 7 | 27.80 | 0.0001 | 78% |
| People with diseases | 8 | 26.21 | 0.0005 | 73% |
| **Age** |  |  |  |  |
| 18–45 | 2 | 3.85 | 0.05 | 74% |
| 45–60 | 8 | 57.21 | <0.00001 | 88% |
| ≥60 | 4 | 55.32 | <0.00001 | 95% |
| **BMI (Kg/m$^2$)** |  |  |  |  |
| ≤24 | 2 | 11.48 | 0.0007 | 91% |
| >24 | 8 | 103.29 | <0.00001 | 93% |
| **TC (mmol/L)** |  |  |  |  |
| <5.2 | 7 | 19.12 | 0.004 | 69% |
| ≥5.2 | 5 | 55.69 | <0.00001 | 93% |
| **SBP (mmhg)** |  |  |  |  |
| <140 | 7 | 40.95 | <0.00001 | 85% |
| ≥140 | 3 | 31.97 | <0.00001 | 94% |
| **DBP (mmhg)** |  |  |  |  |
| <90 | 8 | 104.84 | <0.00001 | 93% |
| ≥90 | 1 | - | - | - |
| **Triglyceride(mmol/L)** |  |  |  |  |
| <1.7 | 9 | 45 | <0.00001 | 82% |
| ≥1.7 | 1 | - | - | - |
| **LDL-C(mmol/L)** |  |  |  |  |
| <3.4 | 8 | 109.02 | <0.00001 | 94% |
| ≥3.4 | 1 | - | - | - |

Note: chi$^2$ represents the chi square value in the Q test.

population differences, the population was divided into groups according to the disease situation through subgroup analysis. We found that increased uric acid levels were associated with carotid intimal thickening in both healthy individuals and those at high risk for cardiovascular diseases such as type 2 diabetes, hypertension, and metabolic syndrome. In addition, subgroup analysis of other risk factors for cardiovascular and cerebrovascular diseases, including age, BMI, TC, SBP, DBP, LDL-C and triglycerides, found that the correlation between SUA and CIMT was affected by other factors, but the correlation was still established. Therefore, this

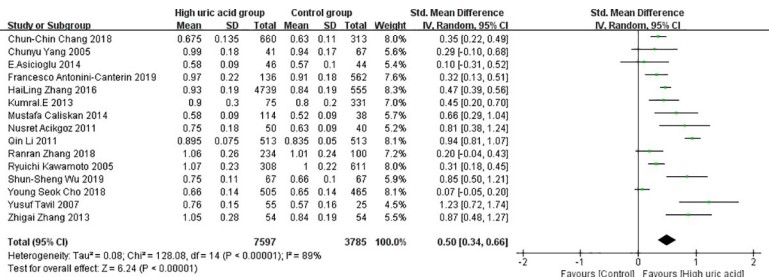

**Fig 3. Overall forest plot of the meta-analysis on the association between SUA and CIMT.**

**Population prevalence:**

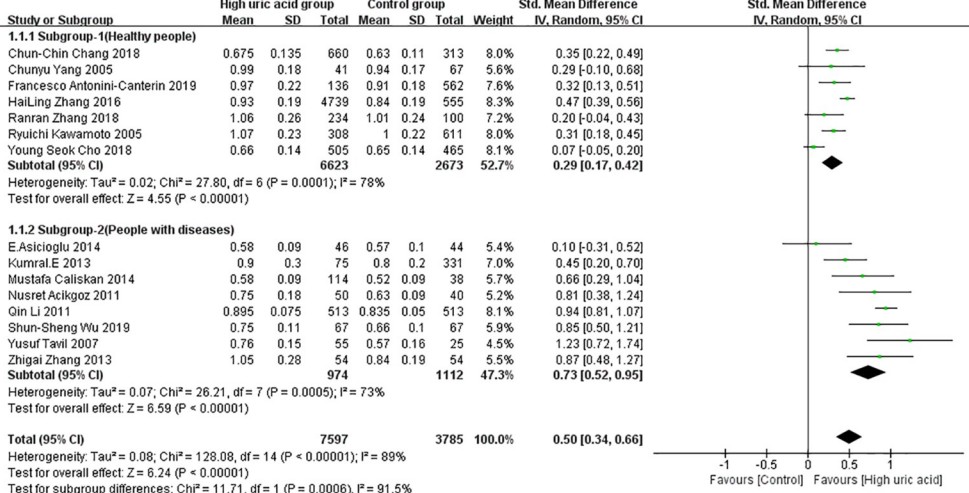

**Age:**

**BMI:**

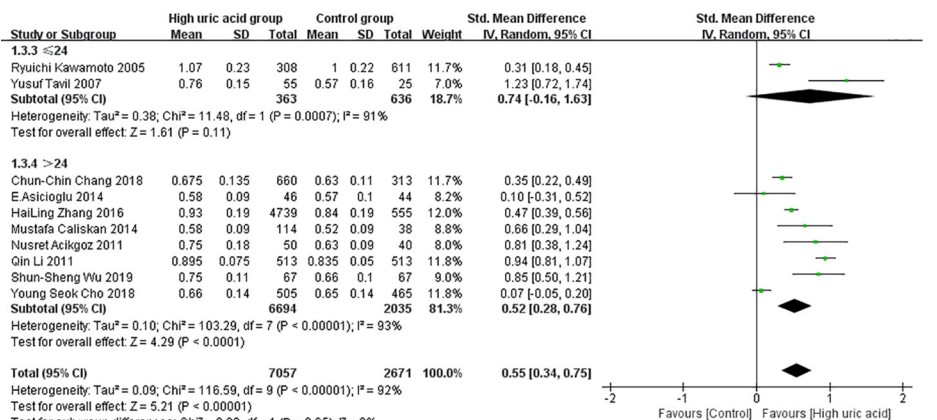

**Fig 4. Subgroup analysis of forest plot based on population prevalence, age, and BMI.**

study revealed that a high level of SUA may be a predictor of atherosclerosis, and strengthening the control of serum uric acid levels in early prevention is helpful to reduce the risk of atherosclerosis. These findings proved that the rising level of SUA could pose an increased risk of cardiovascular and cerebrovascular diseases, provided a potential therapeutic target and new ideas for further clinical researches. It was suggested that decreasing level of uric acid could be an adjunctive therapy for cardiovascular and cerebrovascular diseases, and the atherosclerosis diagnosis and treatment also could be supported by testing the level of SUA in clinic.

There was strong heterogeneity among the 15 articles. Considering that there are differences in the correlation between SUA and CIMT among different populations and combined with the forest plot, it is preliminarily estimated that the heterogeneity comes from the study population. By removing each study one by one and calculating the heterogeneity among the remaining studies, we found that the population differences were likely to be the cause of the high heterogeneity. In the 6 studies by Zhigai Zhang 2013 [22], Qin Li 2011 [25], Mustafa Caliskan 2017 [29], Nusret Acikgoz 2011 [31], Yusuf Tavil 2007 [32], and Shun-Sheng Wu 2019 [33], study subjects had type 2 diabetes, hypertension, cardiac syndrome X, and metabolic syndrome (no renal disease or disability), and heterogeneity disappeared when subgroup analysis was performed in these six studies.

There are some limitations in this study. First, the study failed to explore differences in correlation between men and women because most of the included literature failed to group by sex. Second, similarly, different conditions of CIMT (thickened, abnormal and plaque, etc.) were not distinguished in the included literature, resulting in the inability to conduct subgroup analysis on the correlation between SUA and different types of CIMT abnormalities. Third, the analysis was also limited by the number of included studies and the quality of individual studies. We were unable to conduct an adequate combinatorial analysis during the subgroup analysis, and many other important factors affecting CIMT could not be analysed. Finally, unmeasured and residual confounding may result in confusion in relation to the conclusion. Given the limitations of our meta-analysis, further large-scale studies and sufficient samples

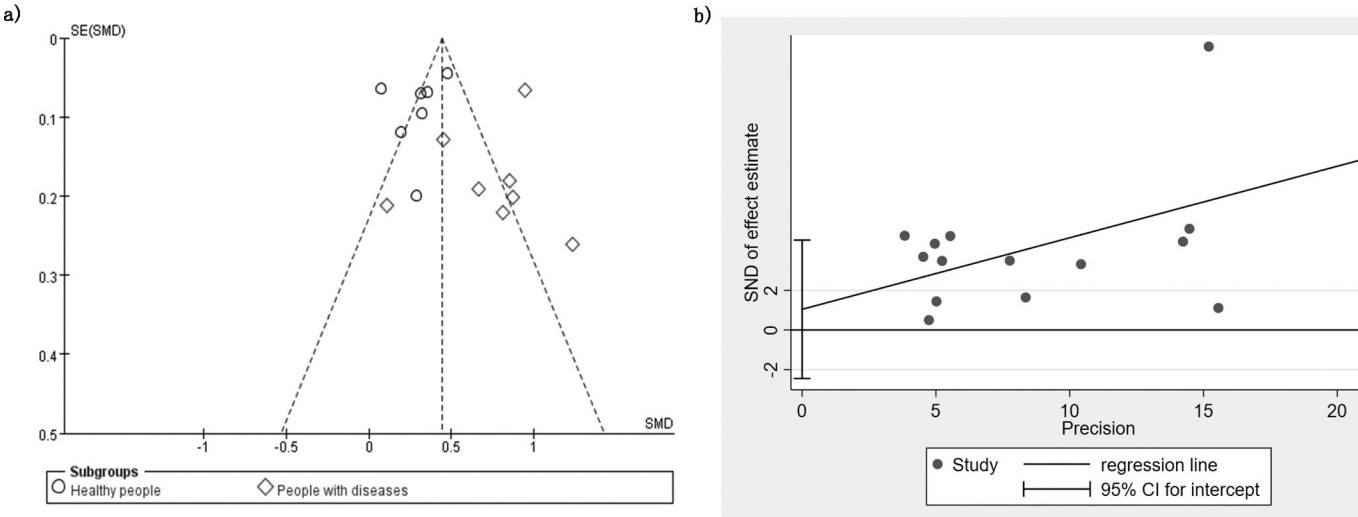

**Fig 5. Publication bias analysis.**

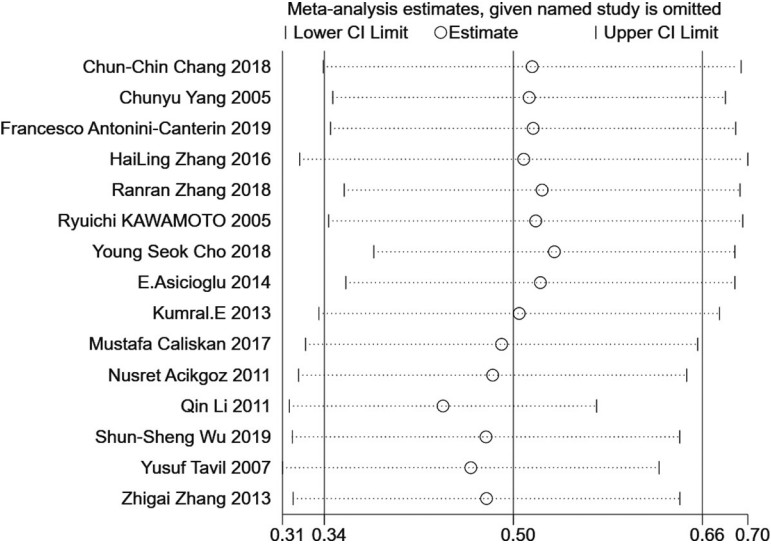

**Fig 6. Sensitivity analysis.**

are needed to demonstrate a convincing link between serum uric acid level and carotid intima-media thickness.

## Conclusion

The results of the meta-analysis show that there is a relationship between serum uric acid level and carotid intima-media thickness in both healthy and diseased people, but the findings cannot show whether this is a causal relationship. At present, the research in this field mainly includes molecular and cell biology research and cross-sectional observational exploration. The conclusion of this study has great guiding significance for related research on risk factors for carotid intimal thickness thickening, suggesting that in the prevention of atherosclerosis, in addition to traditional elements, attention should be paid to the prevention and treatment of hyperuricaemia. The level of serum uric acid can be taken as an important reference basis. In clinical practice, early detection of serum uric acid should be considered, and an appropriate reduction in the serum uric acid level is necessary to prevent carotid intimal thickening. On this basis, cardio-cerebrovascular disease experts should advise their patients to consume a reasonable diet, develop good lifestyle habits, and prevent and avoid the formation of hyperuricaemia and carotid atherosclerosis.

## Supporting information

**S1 Appendix. Subgroup analysis of forest plots based on other risk factors for cardiovascular and cerebrovascular diseases.**
(DOCX)

**S2 Appendix. PRISMA checklist.**
(DOC)

**S1 Table. Summary of the included literature characteristics and key findings.**
(DOCX)

**S2 Table. Confounding factors in the 15 references included in the meta-analysis.**
(DOCX)

## Acknowledgments

We are grateful for all professors and colleagues for helping us during the current research.

## Author Contributions

**Conceptualization:** Mingzhu Ma, Liangxu Wang, Wenjing Huang, Xiaoni Zhong, Huan Wang, Bin Peng, Min Mao.

**Data curation:** Mingzhu Ma, Liangxu Wang, Wenjing Huang.

**Formal analysis:** Mingzhu Ma, Liangxu Wang, Wenjing Huang, Huan Wang.

**Investigation:** Min Mao.

**Methodology:** Mingzhu Ma, Liangxu Wang, Wenjing Huang, Xiaoni Zhong, Longfei Li, Huan Wang, Bin Peng, Min Mao.

**Resources:** Min Mao.

**Software:** Mingzhu Ma, Liangxu Wang, Wenjing Huang, Longfei Li, Bin Peng.

**Supervision:** Xiaoni Zhong, Min Mao.

**Validation:** Mingzhu Ma, Liangxu Wang, Min Mao.

**Visualization:** Mingzhu Ma, Liangxu Wang.

**Writing – original draft:** Mingzhu Ma, Liangxu Wang.

**Writing – review & editing:** Mingzhu Ma, Liangxu Wang, Xiaoni Zhong, Longfei Li, Huan Wang, Bin Peng, Min Mao.

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
