## [Decision Letter · Decision Letter 0]

6 Nov 2020

PONE-D-20-28226

Meta-analysis of the correlaion between serum uric acid level and carotid intima-media thickness

PLOS ONE

Dear Dr. Mao,

Thank you for submitting your manuscript to PLOS ONE. After careful consideration, we feel that it has merit but does not fully meet PLOS ONE’s publication criteria as it currently stands. Therefore, we invite you to submit a revised version of the manuscript that addresses the points raised during the review process.

We look forward to receiving your revised manuscript.

Kind regards,

Giuseppina Novo

Academic Editor

PLOS ONE

Journal Requirements:

2. Please provide the following information in your Methods section relating to the search terms used for the systematic review. Please also amend the PRISMA checklist to display the corresponding page numbers of each checklist item.

3. Thank you for stating the following financial disclosure: [NO STATEMENT PROVIDED]

*Please include your amended statements within your cover letter; we will change the online submission form on your behalf.*

4. Thank you for stating the following in your Competing Interests section: [NO STATEMENT PROVIDED]

A. Please complete your Competing Interests statement to state any Competing Interests. If you have no competing interests, please state "The authors have declared that no competing interests exist.", as detailed online in our guide for authors at http://journals.plos.org/plosone/s/submit-now

5. Please amend either the title on the online submission form (via Edit Submission) or the title in the manuscript so that they are identical.

6. Please include captions for your Supporting Information files at the end of your manuscript, and update any in-text citations to match accordingly. Please see our Supporting Information guidelines for more information: http://journals.plos.org/plosone/s/supporting-information

Reviewers' comments:

Reviewer's Responses to Questions

**Comments to the Author**

1. Is the manuscript technically sound, and do the data support the conclusions?

Reviewer #1: Partly

Reviewer #2: Yes

Reviewer #3: Partly

2. Has the statistical analysis been performed appropriately and rigorously? 

Reviewer #1: I Don't Know

Reviewer #2: Yes

Reviewer #3: No

3. Have the authors made all data underlying the findings in their manuscript fully available?

Reviewer #1: Yes

Reviewer #2: Yes

Reviewer #3: Yes

4. Is the manuscript presented in an intelligible fashion and written in standard English?

Reviewer #1: No

Reviewer #2: Yes

Reviewer #3: No

5. Review Comments to the Author

Reviewer #1: I think I made a mistake accepting the revision of this article whose validity is substantially linked to the soundness of its statistical approach that goes, unfortunately, beyond my level knowledge. Topic is certainly interesting but, given the ambiguity about this subject, as reported in the literature, I think that a sound statistical check (that goes beyond my capacity) is necessary.

There are a few grammatical errors in the text. The style of references is not uniform across the list

Reviewer #2: In this paper, authors analyze through meta-analysis the relationship between serum uric acid level and carotid

intimal thickness.The paper is well designed and the conclusion, such as that there is a significant correlation between serum uric acid level and carotid intima-media thickness, and a high concentration of serum uric acid is related to carotid artery intima-media thickness are beliavable.

I would suggest some revisions as follows:

- in the abstract, I suggest to better explicate these sentence that was repeated: "Meta-analysis showed that carotid intimal thickness in the high uric acid group was significantly higher than that in the control group [...] The results showed that the carotid intimal thickness of the high uric acid group was still significantly higher than that of the control group, and the difference was statistically significant".

- the 15 articles selected for the meta-analysis have not been cited; please add the references of these articles to allow readers to know them, maybe it would be added in the Table 1;

- Figure 5 contains a mistake in the author "Francesco Antonini-Canterin" that was wrongly written.

Reviewer #3: This is a meta-analysis study focus on the relationship between serum uric acid level the carotid intima-media thickness. Previous studies have shown that CIMT is the main risk factor and main pathophysiological basis of ASCVD. However, the correlation between hyperuricemia and ASCVD is still controversial. In current study contain some serious flaws and some concerns need to be addressed,

1. CIMT determination uses different B-ultrasound equipment in different research centers, and uses different operating procedures, how to evaluate the differences between centers?

2. The population included in the study is biased, and there are high-risk groups of ASCVD such as diabetes, which requires further subgroup analysis.

3. Other influencing factors of ASCVD such as blood lipid level, BP, medication status, BMI, etc. may be alternative in different studies. This will ultimately affect the value of CMIT and should be included in the analysis.

4. The mechanism of the effect of hyperuricemia on CIMT and even atherosclerosis is not sufficiently explained in the discussion section, and further elucidation is needed.

5. The normal value of CMIT is generally considered to be below 1.0mm, 1.0-1.2mm is considered to be thickened, and 1.2mm or more is considered abnormal. Can the CIMT value be subdivided into different subgroups for analysis and discuss the relationship between hyperuricemia and the abnormal rate of CMIT?

6. Professional language editing must be done prior to re-submission.

6. PLOS authors have the option to publish the peer review history of their article (what does this mean?). If published, this will include your full peer review and any attached files.

Reviewer #1: No

Reviewer #2: **Yes: **Concetta Di Nora

Reviewer #3: No

---

## [Author Response · Author response to Decision Letter 0]

9 Dec 2020

Editor:

*Any new text in the revised manuscript has been noted below in bold font.

1. In accordance with PLOS ONE’s style requirements, we have updated the file names and manuscript title, as well as the references.

2. Search terms used for the systematic review have been added to the Methods section, and PRISMA checklist has been updated in accordance with PLOS guidelines.

3. Statements of funding information have been included at the cover letter.

4. Description of competing Interests has been included at the cover letter.

5. The title in the manuscript has been modified according to the title of the online submission system.

6. Supporting Information files have been included at the end of the manuscript, and in-text citations have been updated in accordance with PLOS guidelines.

Reviewer #1:

We thank Reviewer #1 for the significant feedback. We strongly agree with the reviewer's opinion that a statistical check should be conducted on ambiguous subjects, which is also the purpose of our meta-analysis: we conducted a comprehensive analysis of the existing research with a statistical review to expand the sample size and better explore the correlation between serum uric acid level and carotid intimal thickness. 

The revised article was submitted after careful language review, and the style of references has been uniformly modified in accordance with PLOS guidelines. Thanks for your correction!

Reviewer #2:

We thank Reviewer #2 for the important suggestion to explicate the sentence that was repeated. The second half of the repeated part in the abstract is actually the conclusion of the subgroup analysis. We have made the following corrections (Abstract, lines 17-26):

“Meta-analysis showed that CIMT in the high uric acid group was significantly higher than that in the control group (SMD = 0.53, 95% CI: [0.38, 0.68]), and the difference was significant (z = 6.98, P < 0.00001). The heterogeneity among the 15 articles was obvious (I2 = 89%, P < 0.00001). Subgroup analysis by disease status illustrated a positive relationship between SUA and CIMT in healthy people and people with diseases. SUA was shown to be positively correlated with CIMT in people aged 45-60 years and ≥60 years by subgroup analysis by age. SUA was also found to be positively correlated with CIMT in a population with BMI>24 kg/m2 by subgroup analysis by BMI. In addition, subgroup analysis of other risk factors for CIMT, including TC, SBP, DBP, triglycerides, and LDL-C, all showed a positive correlation between SUA and CIMT.”

We thank Reviewer #2 for the suggestion to add the 15 articles selected for the meta-analysis to the references. We have added these 15 articles to the reference section ([20-34]) and added the "Author" column in Table 1, with references to the corresponding literature after the author's name.

The mistake in Figure 5 has been corrected.

Reviewer #3:

We thank reviewer #3 for the extensive feedback, and we have incorporated most of it. Based on that, we think our paper has improved to a great extent. It also became more comprehensible for the reader. Thank you.

1. We would like to thank the Reviewer for raising the question of different instruments and procedures for measuring CIMT between studies. In the process of measurement, random error is inevitable due to different operators and instruments. However, the measurement methods of each research centre are strictly in accordance with the medical testing guidelines and consensus [1-4] and control the error within the acceptable range. Thus, the difference is subtle.

2. We agree with the Reviewer that further subgroup analysis is necessary. We conducted a subgroup analysis between healthy people and people at high risk of cardiovascular and cerebrovascular disease. The results are shown in Table 2, and the heterogeneity of subgroups is shown in Table 3. The following text was added to the Results section (Results, lines 156-164):

“Because the correlation between SUA and CIMT is affected by population characteristics and may be affected by other risk factors for CIMT, we conducted subgroup analyses. Initially, we divided the population into two groups and found that CIMT levels were higher in the high uric acid group than in the control group, both in healthy people and in people with diseases. (Healthy people: SMD = 0.29, 95% CI: [0.17, 0.42]; people with diseases: SMD = 0.73, 95% CI: [0.52, 0.95]). The differences were statistically significant (P < 0.00001), as shown in Fig 4, which meant that SUA levels were positively correlated with CIMT. In addition, the heterogeneity of the two subgroups decreased but was still significant (I2 = 78%, P = 0.0001; I2 = 73%, P = 0.0005).”

We also added the results of subgroup analysis to the Abstract (Abstract, lines 20-21):

“Subgroup analysis by disease status illustrated a positive relationship between SUA and CIMT in healthy people and people with diseases.”

3. We strongly agree with the opinion that confounding factors should be considered when researching the correlation between SUA and CIMT. We have added some subgroup analyses of cardiovascular and cerebrovascular disease risk factors that may influence CIMT thickness (age, BMI, TC, SBP, DBP, LDL-C and triglyceride) and added the following content to the Results section (Results, lines 165-173):

“To further control for confounders, we performed an age-based subgroup analysis and found a positive correlation between SUA and CIMT in people aged 45-60 years and ≥60 years (Fig 4), but no correlation was found in people between the ages of 18 and 45. We also performed a subgroup analysis based on BMI. The SUA level was significantly positively correlated with CIMT in people with BMI>24 kg/m2. In addition, we performed subgroup analysis on other risk factors for CIMT: TC, SBP, DBP, triglycerides and LDL-C (the forest plots are shown in S1 Appendix). The results showed that the carotid intimal thickness of the high uric acid group was higher than that of the control group, and the difference was statistically significant (Table 2). The heterogeneity of each subgroup is shown in Table 3.”

We also added the results of subgroup analysis to the Abstract (Abstract, lines 22-26):

“SUA was shown to be positively correlated with CIMT in people aged 45-60 years and ≥60 years by subgroup analysis by age. SUA was also found to be positively correlated with CIMT in a population with BMI>24 kg/m2 by subgroup analysis by BMI. In addition, subgroup analysis of other risk factors for CIMT, including TC, SBP, DBP, triglycerides, and LDL-C, all showed a positive correlation between SUA and CIMT.”

However, because the number of included studies was limited and the confounders included in each study were not exactly the same (we added the confounders of each article to S2 Table), some confounders were included in too few articles. We were unable to conduct an adequate combinatorial analysis during the subgroup analysis, nor were we able to analyse many other important factors that might affect CIMT, which may result in confusion in relation to the conclusion. This is also one of the limitations of our research. We have added this information to the limitations of the article (Discussion, lines 276-279):

“Third, the analysis was also limited by the number of included studies and the quality of individual studies. We were unable to conduct an adequate combinatorial analysis during the subgroup analysis, and many other important factors affecting CIMT could not be analysed.”

4. We appreciate the suggestion by Reviewer #3 to further discuss the mechanisms of hyperuricaemia in CIMT and atherosclerosis. We have added to the Discussion section a description of how hyperuricaemia affects CIMT and atherosclerosis (Discussion, lines, 204-223):

“The destruction of the vascular wall and endothelial cell function are direct causes of atherosclerosis. As an end product of purine metabolism in humans and higher primates, uric acid can damage the integrity of the vascular surface and affect the function of endothelial cells through direct or indirect effects on the vascular endothelium. The mechanism of this process is as follows: uric acid can promote the proliferation and migration of vascular smooth muscle cells and even lead to vascular remodelling, affecting its normal function. When the concentration of uric acid in the blood reaches a certain level, it will deposit into the tissue in the form of urate crystals and damage the vascular endothelium. The occurrence of hyperuricaemia aggravates abnormalities in metabolites and affects the vascular endothelium and haemodynamics [23]. Oxidative stress is the key link in endothelial dysfunction caused by hyperuricaemia and is related to the increased risk of cardiovascular and cerebrovascular disease [39]. However, there is another effect of uric acid level on carotid atherosclerosis. When blood vessels are in the atherosclerotic state, the pro-oxidant state will inactivate some superoxide dismutases (e.g., NOS1 and NOS3, etc.) and reduce the scavenging of free radicals in blood vessels, thus increasing the oxygen level [40]. Uric acid protects the activity of these inactivated enzymes and may counteract the harmful effects of hyperuricaemia in some areas of the vascular system [26]. Because uric acid is generally considered to have more harmful than protective effects, it is considered to be a risk factor for CIMT and atherosclerosis [36, 37, 42], but the clinical relevance of the two has been controversial.”

5. We thank Reviewer #3 very much for posing this important question, which is what we want to solve. However, most of the literature measured only CIMT thickness and did not classify the population according to the status of CIMT (thickened, abnormal or plaque), so we failed to explore the relationship between hyperuricaemia and the abnormal rate of CIMT through subgroup analysis. We have added the following to illustrate these limitations in the discussion section (Discussion, lines 273-276):

“Second, similarly, different conditions of CIMT (thickened, abnormal and plaque, etc.) were not distinguished in the included literature, resulting in the inability to conduct subgroup analysis on the correlation between SUA and different types of CIMT abnormalities.”

6. Professional language editing has been completed.

Reference:

1. Management of Atherosclerotic Carotid and Vertebral Disease: 2017 Clinical Practice Guidelines of the European Society for Vascular Surgery (ESVS):http://sr.xingshulin.com/view/guide.html?id=13703&share_ticket=code%3D-1%26version%3D7.16.0%26os%3Da

2. Saba L, Yuan C, Hatsukami TS, Balu N, Qiao Y, DeMarco JK, et al. Carotid Artery Wall Imaging: Perspective and Guidelines from the ASNR Vessel Wall Imaging Study Group and Expert Consensus Recommendations of the American Society of Neuroradiology. AJNR Am J Neuroradiol. 2018;39(2):E9-e31. Epub 2018/01/13. doi: 10.3174/ajnr.A5488. PubMed PMID: 29326139; PubMed Central PMCID: PMCPMC7410574.

3. Xiaoying Li, Heng Guan, Tingshu Yang, Wei Guo, Shuxia Wang. Chinese experts suggest the diagnosis and treatment of atherosclerotic diseases in the limbs of the elderly (2012). Chinese Journal of Geriatrics. 2013;(02):121-31.(in Chinese)

4. Chinese guidelines for interventional diagnosis and treatment of carotid artery stenosis: http://sr.xingshulin.com/view/guide.html?id=7150&share_ticket=code%3D-1%26version%3D7.16.0%26os%3Da

---

## [Decision Letter · Decision Letter 1]

8 Jan 2021

PONE-D-20-28226R1

Meta-analysis of the correlation between serum uric acid level and carotid intima-media thickness

PLOS ONE

Dear Dr. Mao,

Thank you for submitting your manuscript to PLOS ONE. After careful consideration, we feel that it has merit but does not fully meet PLOS ONE’s publication criteria as it currently stands. Therefore, we invite you to submit a revised version of the manuscript that addresses the points raised during the review process.

Particularly revise the discussion according to one of the reviewers suggestions. Afterwards there are no more comments to be addressed.

We look forward to receiving your revised manuscript.

Kind regards,

Giuseppina Novo

Academic Editor

PLOS ONE

Reviewers' comments:

Reviewer's Responses to Questions

**Comments to the Author**

1. If the authors have adequately addressed your comments raised in a previous round of review and you feel that this manuscript is now acceptable for publication, you may indicate that here to bypass the “Comments to the Author” section, enter your conflict of interest statement in the “Confidential to Editor” section, and submit your "Accept" recommendation.

Reviewer #2: All comments have been addressed

Reviewer #3: All comments have been addressed

2. Is the manuscript technically sound, and do the data support the conclusions?

Reviewer #2: Yes

Reviewer #3: Yes

3. Has the statistical analysis been performed appropriately and rigorously? 

Reviewer #2: Yes

Reviewer #3: Yes

4. Have the authors made all data underlying the findings in their manuscript fully available?

Reviewer #2: Yes

Reviewer #3: Yes

5. Is the manuscript presented in an intelligible fashion and written in standard English?

Reviewer #2: Yes

Reviewer #3: Yes

6. Review Comments to the Author

Reviewer #2: In this paper, authors analyze through meta-analysis the relationship between serum uric acid level and carotid intimal thickness. The paper is well designed and the conclusion, such as that there is a significant correlation

between serum uric acid level and carotid intima-media thickness, and a high concentration of serum uric acid is related to carotid artery intima-media thickness are beliavable.

All the previous concerns have been adequately addressed by the authors.

Reviewer #3: The discussion could be more concise and highlight its clinical significance, in addition, there is a little no more revision needed.

7. PLOS authors have the option to publish the peer review history of their article (what does this mean?). If published, this will include your full peer review and any attached files.

Reviewer #2: **Yes: **Concetta Di Nora

Reviewer #3: No

---

## [Author Response · Author response to Decision Letter 1]

13 Jan 2021

Editor:

*Any new text in the revised manuscript has been noted below in bold font.

The Discussion section has been modified according to the suggestions of reviewers.

RESPONSE TO REVIEWER’S COMMENTS:

Reviewer #3: The discussion could be more concise and highlight its clinical significance, in addition, there is a little no more revision needed.

Reply: We appreciate the excellent observation from Reviewer #3 regarding the Discussion section. We adopted the suggestions of reviewers, revised the discussion and added the following contents to highlight the clinical significance (Discussion, lines 244-251):

“Therefore, this study revealed that a high level of SUA may be a predictor of atherosclerosis, and strengthening the control of serum uric acid levels in early prevention is helpful to reduce the risk of atherosclerosis. These findings proved that the rising level of SUA could pose an increased risk of cardiovascular and cerebrovascular diseases, provided a potential therapeutic target and new ideas for further clinical researches. It was suggested that decreasing level of uric acid could be an adjunctive therapy for cardiovascular and cerebrovascular diseases, and the atherosclerosis diagnosis and treatment also could be supported by testing the level of SUA in clinic.”

---

## [Decision Letter · Decision Letter 2]

20 Jan 2021

Meta-analysis of the correlation between serum uric acid level and carotid intima-media thickness

PONE-D-20-28226R2

Dear Dr. Mao,

We’re pleased to inform you that your manuscript has been judged scientifically suitable for publication and will be formally accepted for publication once it meets all outstanding technical requirements.

Kind regards,

Giuseppina Novo

Academic Editor

PLOS ONE

Additional Editor Comments (optional):

Reviewers' comments:

Reviewer's Responses to Questions

**Comments to the Author**

1. If the authors have adequately addressed your comments raised in a previous round of review and you feel that this manuscript is now acceptable for publication, you may indicate that here to bypass the “Comments to the Author” section, enter your conflict of interest statement in the “Confidential to Editor” section, and submit your "Accept" recommendation.

Reviewer #2: All comments have been addressed

Reviewer #3: All comments have been addressed

2. Is the manuscript technically sound, and do the data support the conclusions?

Reviewer #2: (No Response)

Reviewer #3: Yes

3. Has the statistical analysis been performed appropriately and rigorously? 

Reviewer #2: (No Response)

Reviewer #3: N/A

4. Have the authors made all data underlying the findings in their manuscript fully available?

Reviewer #2: (No Response)

Reviewer #3: Yes

5. Is the manuscript presented in an intelligible fashion and written in standard English?

Reviewer #2: (No Response)

Reviewer #3: Yes

6. Review Comments to the Author

Reviewer #2: ----------------------------------------------------------------------------------------------------

Reviewer #3: The author has rigorously answered the reviewer's questions and made corresponding modifications. Now it could be accepted for publication.

7. PLOS authors have the option to publish the peer review history of their article (what does this mean?). If published, this will include your full peer review and any attached files.

Reviewer #2: **Yes: **CONCETTA DI NORA

Reviewer #3: No

---

## [Editor Report · Acceptance letter]

22 Jan 2021

PONE-D-20-28226R2 

Meta-analysis of the correlation between serum uric acid level and carotid intima-media thickness 

Dear Dr. Mao:

I'm pleased to inform you that your manuscript has been deemed suitable for publication in PLOS ONE. Congratulations! Your manuscript is now with our production department. 

Kind regards, 

on behalf of

Professor Giuseppina Novo 

Academic Editor

PLOS ONE